# Leukoaraiosis as a Promising Biomarker of Stroke Recurrence among Stroke Survivors: A Systematic Review

Theofanis Dimaras [1,†], Ermis Merkouris [1,†], Dimitrios Tsiptsios [1,*], Foteini Christidi [1], Anastasia Sousanidou [1], Ilias Orgianelis [1], Efthymia Polatidou [1], Iordanis Kamenidis [1], Stella Karatzetzou [1], Aimilios Gkantzios [1], Christos Ntatsis [1], Christos Kokkotis [2], Sofia Retsidou [1], Maria Aristidou [2], Maria Karageorgopoulou [2], Evlampia A. Psatha [1], Nikolaos Aggelousis [2] and Konstantinos Vadikolias [1]

1   Neurology Department, Democritus University of Thrace, 68100 Alexandroupolis, Greece; thiou295@gmail.com (T.D.); ermimerk@med.duth.gr (E.M.); christidi.f.a@gmail.com (F.C.); anastasiasousanidou@gmail.com (A.S.); evthpola@med.duth.gr (E.P.); iordkame1@med.duth.gr (I.K.); skaratzetzou@gmail.com (S.K.); aimilios.gk@gmail.com (A.G.); chrintat@med.duth.gr (C.N.); sofirets@med.duth.gr (S.R.); eviepsatha@yahoo.gr (E.A.P.); vadikosm@yahoo.com (K.V.)
2   Department of Physical Education and Sport Science, Democritus University of Thrace, 69100 Komotini, Greece; ckokkoti@affil.duth.gr (C.K.); maariste@phyed.duth.gr (M.A.); mkarageo@phyed.duth.gr (M.K.); nagelous@phyed.duth.gr (N.A.)
*   Correspondence: tsiptsios.dimitrios@yahoo.gr; Tel.: +30-694-432-0016
†   These authors contributed equally to this work.

**Abstract:** Stroke is the leading cause of functional disability worldwide, with increasing prevalence in adults. Given the considerable negative impact on patients' quality of life and the financial burden on their families and society, it is essential to provide stroke survivors with a timely and reliable prognosis of stroke recurrence. Leukoaraiosis (LA) is a common neuroimaging feature of cerebral small-vessel disease. By researching the literature of two different databases (MEDLINE and Scopus), the present study aims to review all relevant studies from the last decade, dealing with the clinical utility of pre-existing LA as a prognostic factor for stroke recurrence in stroke survivors. Nineteen full-text articles published in English were identified and included in the present review, with data collected from a total of 34,546 stroke patients. A higher rate of extended LA was strongly associated with stroke recurrence in all stroke subtypes, even after adjustment for clinical risk factors. In particular, patients with ischemic stroke or transient ischemic attack with advanced LA had a significantly higher risk of future ischemic stroke, whereas patients with previous intracerebral hemorrhage and severe LA had a more than 2.5-fold increased risk of recurrent ischemic stroke and a more than 30-fold increased risk of hemorrhagic stroke. Finally, in patients receiving anticoagulant treatment for AF, the presence of LA was associated with an increased risk of recurrent ischemic stroke and intracranial hemorrhage. Because of this valuable predictive information, evaluating LA could significantly expand our knowledge of stroke patients and thereby improve overall stroke care.

**Keywords:** leukoaraiosis; white matter hyperintensities; stroke; stroke recurrence; prognosis



## 1. Introduction

Stroke is a significant global health challenge worldwide; it is the second leading cause of death, following heart disease, and the primary cause of acquired disability in adults [1,2]. Moreover, a recent systematic analysis considering data from the last three decades shows that the burden of stroke is steadily increasing, as the absolute number of cases has increased by 70% compared to 1990 [3]. With more than 50% of stroke patients over the age of 65, the continued growth of the aging population is likely to lead to a substantial increase in the number of stroke survivors [4]. Furthermore, the number of people over 60 years of age in affluent countries is expected to triple by 2050 [5]—a fact that further underscores the need for the prompt and accurate identification of patients

with an unfavorable prognosis. As a result, this can facilitate personalized rehabilitation programs tailored to each individual's recovery potential and promote long-term functional independence for stroke survivors. Regarding the epidemiology of stroke recurrence, the probability varies depending on several factors. The type and cause of the initial stroke have been demonstrated to play a significant role in the chance of recurrence, with small-vessel occlusion and ischemic stroke being at the lowest end of the spectrum of probability and cardioembolic stroke being at the highest end [6]. The individual's age also has a big impact on the predisposition to another vascular event, as was elucidated by Modrego et al. [7], who found a positive correlation between an individual's age and the chance of recurrence. Among other risk factors that show a clear correlation are diseases that disrupt the vascular endothelium such as diabetes [8] and high blood pressure [9]. There are also lifestyle changes that can have a significant impact on the individual's propensity to another stroke, such as smoking [10] and heavy alcohol use [11]. About one in four stroke survivors will experience another stroke within five years [12] Therefore, the ability to forecast which group of stroke survivors has the highest probability of recurrence is monumental for timely intervention. The need to develop biomarkers is evident and several candidates showing promise in aiding as prognostic factors in an acute ischemic stroke settings were examined in other systematic reviews [13].

According to the Biomarkers Definitions Working Group, the term biomarker refers to "a characteristic that is objectively measured and evaluated as an indication of normal biological processes, pathogenic processes, or pharmacological responses to a therapeutic intervention" [14]. However, not everything contained in that statement can be considered an ideal biomarker. Several other factors must be fulfilled to achieve a truly useful and timeless biomarker. First, it needs to have high specificity and sensitivity while also being easy to obtain and reproduce from the available sources. Another prerequisite is the ability to form a picture of the underlying pathophysiology of the endovascular cascade to prove clinical relevance [15].

Up to today, several such biomarkers have been investigated. Some aim to depict the state of the microvascular health of the brain and the stress that the body is under to predict the probability of stroke recurrence. A classic example of this class of biomarkers is the copeptin levels that have been demonstrated to correlate with stroke recurrence [16]. Others may point to systemic inflammation, which increases the rate of stroke recurrence through various mechanisms such as inducing a hypercoagulable state or causing endothelial damage. This category includes Interleukin-6 (IL-6) and YKL-40, which were demonstrated in a separate study to be linked with stroke recurrence [17]. Another promising biomarker for stroke recurrence appears to be the amount of lipoprotein A detected in the blood of the patient—a correlation that is, however, attenuated when LDL-c and inflammation markers remain low [18]. There are also neuroimaging biomarkers that provide an indirect look at the brain reserve by examining the lesions that are present in cases of deterioration and the loss of healthy brain tissue. Such biomarkers are cerebral microbleeds, which have been correlated with stroke recurrence in a different study [19], and leukoaraiosis (LA), with the latter being evaluated further in this review for its ability to predict which individuals have an increased probability of stroke recurrence.

LA, also known as white matter lesions (WMLs), is a neuroimaging phenomenon that is most commonly observed in elderly people and refers to specific abnormalities of white matter (WM), which typically present as either multifocal or diffuse changes of varying sizes that are predominantly found within the periventricular space [20]. Hachinski and his colleagues originally reported the aforementioned lesioned regions in 1987 as both symmetrical and bilateral WM alterations in the vicinity of the cerebral ventricles or inside the semi-oval center observed on brain scans of elderly or demented individuals [21]. To measure the prevalence and severity of WMLs, several methods of assessment have been used. The Fazekas score, which assesses both periventricular and deep WM alterations regarding the quantity and magnitude of lesions and distinguishes between focal or punctate, early confluent, and confluent abnormalities, is the most frequently utilized method [22].

Regarding the pathophysiological framework of LA, although the precise mechanism remains incompletely understood, the development of WM anomalies is considered to be a result of ischemia, after an analysis of morphological, physiopathologic, and clinico-pathologic findings [23]. Selective injuries in the hemispheric WM have been noted in a limited number of human conditions and are also known as leukoencephalopathies; they are characterized by hypoxia/ischemia to the brain from a prolonged reduction of oxygenation and suboptimal circulation in the area. One classic example is carbon monoxide poisoning (Grinker's myelinopathy); although, in this condition, direct carbon monoxide toxicity could contribute to the brain lesion. Similar histological changes of the WM, such as coagulative necrosis and cavitation [24] to nonspecific tissue changes such as sponginess, patchy demyelination, and astrocytic proliferation [25], have been noted in both leukoencephalopathies and LA. It is uncertain why some ischemic injuries selectively damage the WM, but the distinct pattern of blood flow to the WM may be both a risk factor and a localizing factor. The cerebral hemispheric WM obtains the majority of its blood supply via long penetrating arteries coming from the pial network on the brain's surface. These penetrating arteries begin at right angles from the subarachnoid vessels, proceed perpendicular to the brain cortex through the cortical layers, and reach the WM via myelinated fibers [26]. A region of the WM immediately proximal to the walls of the lateral ventricles is supplied with blood by ventriculofugal vessels originating from subependymal arteries; these branches arise either from the choroidal arteries or from terminal branches of the rami striati [27]. The ventriculofugal blood vessels, measuring around 15 mm in length, flow towards the penetrating centripetal vessels that arise from the pial surface (bringing vessels or rami medullares). Communication between the vessels that originate from the surface and the ones branching off the subependymal system is measured to be either scarce [28] or not present [26]. This vascularization pattern suggested by de Reuck [29] was that the periventricular WM has an arterial borderline zone (watershed) that appears to be more vulnerable to injury from systemic or localized reductions in blood flow (CBF). The healthy state of this microvasculature can be compromised by age, blood pressure, diabetes, and smoking, among many other things, as was aforementioned, leading to several changes such as the replacement of the smooth muscle cells by fibro-hyaline material with the thickening of the wall and narrowing of the vascular lumen (arteriolosclerosis) [30–32]. For that reason, the presence of severe LA in a patient suggests an already compromised microvasculature of the WM and leads to our hypothesis that such an individual is more likely to have a recurrent vascular event given the common underlying pathophysiological mechanism these two conditions share.

In this systematic review, we inquired about the ability of the presence of WMLs to predict the probability of a stroke recurring so that it can be used in the future in conjunction with other factors as a reliable source of information. So far, our study group has elucidated a connection between LA and depression and cognitive decline [33], worse prognostic outcome overall [34], and worse results in patients with severe LA undergoing reperfusion therapy [35]. Considering the expected substantial increase in the number of strokes and the need for an accurate prognosis of each person's propensity to another stroke, we examined the potential prognostic role of baseline LA in the context of stroke recurrence by reviewing the literature regarding LA and stroke recurrence in the last decade.

## 2. Materials and Methods

The Preferred Reporting Items for Systematic Reviews and Meta-Analyses (PRISMA) checklist (PRISMA registration number: CRD42023442725) was used to guide this study. Our study's methods were a priori designed.

### 2.1. Search Strategy

Two investigators (TD and EM) conducted literature research on two databases (MEDLINE and Scopus) to trace all relevant studies published between 1 January 2012 and 25 June 2022. Search terms were as follows: ("leukoaraiosis" OR "white matter hyperintensities" OR

"WMHs") AND ("stroke recurrence"). The retrieved articles were also hand-searched for any further potential eligible articles. Any disagreement regarding the screening or selection process was solved by a third investigator (KV) until a consensus was reached.

### 2.2. Selection Criteria

Only full-text original articles published in the English language were included. Secondary analyses, reviews, guidelines, meeting summaries, comments, unpublished abstracts, or studies conducted on animals were excluded. There was no restriction on study design or sample characteristics.

### 2.3. Data Extraction

Data extraction was performed using a predefined data form created in Excel. We recorded the type of stroke, authors, year of publication, number of participants, biomarker, type of study, type of stroke, number, age, and gender of participants, cerebrovascular risk factors, medication, previous stroke, follow-up time, leukoaraiosis/WMH assessment, severity of leukoaraiosis, time of MRI execution/specification of MRI, scale of stroke severity and prognosis/clinical outcome and, finally, the main results of each study.

### 2.4. Data Analysis

No statistical analysis or meta-analysis was performed due to the high heterogeneity among studies. Thus, the data were only descriptively analyzed.

## 3. Results

### 3.1. Database Searches

Overall, 461 records were retrieved from the database search. Duplicates and irrelevant studies were excluded; hence, a total of 77 articles were selected. After screening the full text of the articles, 19 studies were eligible for inclusion (Figure 1).

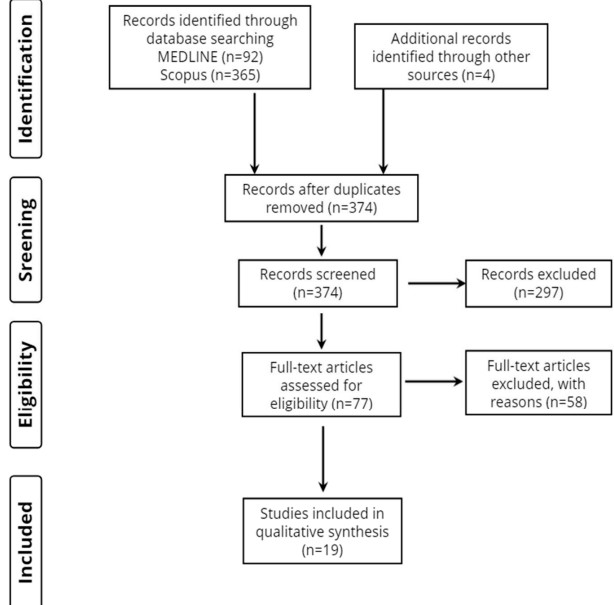

**Figure 1.** Study flow diagram (PRISMA flowchart).

### 3.2. Study Characteristics

A total of 19 publications fulfilled our inclusion criteria, as shown in Table 1. Ten focused entirely on acute ischemic stroke (AIS), four included patients with either AIS or transient ischemic attack (TIA), three enrolled only patients with TIA, and two of them studied patients regardless of stroke type. Considering the origin of the studies, thirteen were from Asia, five came from Europe, and one was from America.

**Table 1.** Basic characteristics of the 19 included studies.

| Authors, Year of Publication | Type of Study | Biomarker | Type of Stroke | Number of Participants/Age of Participants (Years)/Gender (M/F) | Cerebrovascular Risk Factors (*n*) | Medication (*n*) | Previous Stroke (*n*) | Follow-Up Time | Leukoaraiosis /WMH Assessment | Severity of Leukoaraiosis | Time of MRI Execution/Specification of MRI | Scale of Stroke Severity and Prognosis/Clinical Outcome | Main Results |
|---|---|---|---|---|---|---|---|---|---|---|---|---|---|
| | | | | | **Patients with Ischemic Stroke (IS)** | | | | | | | | |
| Andersen et al., 2017 [36] | Longitudinal (prospective) | WMH | AIS (and no AF) | 832/59.6 ± 13.9 483M/349F | Congestive heart failure (9) Hypertension (396) Diabetes (76) Vascular disease (74) Previous myocardial infarction (21) Peripheral arterial disease (38) Other heart disease (32) Smoking (505) | Antiplatelet treatment (784) Oral anticoagulant treatment (0) | 46 | 3.3 years | Fazekas scale | Moderate to severe | Within 4 weeks after stroke/1.5 or 3.0 Tesla MRI scanners | - | White matter hyperintensities were significantly associated with the risk of recurrent ischemic stroke, even after adjusting for clinical risk factors within the $CHA_2DS_2VASc$ score. |
| Kim et al., 2014 [37] | Longitudinal (retrospective) | Leukoaraiosis | AIS | 2378/70 1283M/1095F | Hypertension (1635) Diabetes mellitus (548) Atrial fibrillation (600) Smoking (457) Presence of chronic infarcts on MRI (360) | Antiplatelet (1669) Anticoagulation (884) Antihypertensive (662) Statin (1566) Cerebral revascularization (128) | 337 | 90 days | Fazekas scale | Mild/extensive | Within 72 h of stroke onset/1.5T GE Signa (GE Medical Systems, Milwaukee, WI, USA) or Siemens Sonata (Siemens Medical Solutions, Erlangen, Germany) scanners | NIHSS at admission | Higher total Fazekas score increased the probability of stroke recurrence. Patients with recurrence had a marginally higher rate of extensive LA and significantly higher rate of extensive periventricular LA, when compared with the patients without recurrence, but there was no such difference in extensive subcortical LA. |

**Table 1.** *Cont.*

| Authors, Year of Publication | Type of Study | Biomarker | Type of Stroke | Number of Participants/Age of Participants (Years)/Gender (M/F) | Cerebrovascular Risk Factors (*n*) | Medication (*n*) | Previous Stroke (*n*) | Follow-Up Time | Leukoaraiosis /WMH Assessment | Severity of Leukoaraiosis | Time of MRI Execution/Specification of MRI | Scale of Stroke Severity and Prognosis/Clinical Outcome | Main Results |
|---|---|---|---|---|---|---|---|---|---|---|---|---|---|
| Ryu et al., 2019 [38] | Longitudinal (prospective) | WMH | AIS | 7101/67.9 ± 12.7 4170M/2931F | Hypertension (4835) Diabetes mellitus (2353) Hyperlipidemia (2467) Smoking (1872) Coronary artery disease (574) Atrial fibrillation (1501) | Statins (1121) | 1435 | 1 year | Volumetric | n/a | n/a/1.5T (*n* = 6099) or 3T (*n* = 1002) MRI systems | NIHSS at admission | There was an association between WMH volume load and stroke recurrence, since the highest quartile of WMH volume had a 2.2-fold increased risk of recurrent stroke compared with the lowest quartile, and although this association was stronger for hemorrhagic than for ischemic stroke, the absolute risk of ischemic recurrence remained higher. |
| Park et al., 2019 [39] | Longitudinal (prospective) | WMH | TIA or AIS | 1454/65.9 ± 10.8 904M/550F | Hypertension (1294) Dyslipidemia (626) Diabetes mellitus (465) Coronary heart disease (67) Smoking (644) Family history of stroke (314) | Antihypertensive (1069) Statin (1160) Probucol (732) Cilostazol (719) Aspirin (735) | 1301 | 1.9 years | Fazekas scale | n/a | At baseline/n/a | NIHSS, MMSE | Advanced WMH (Fazekas score 3) was independently associated with an over 2-fold increment in the risk of recurrent stroke and increasing WMH severity (Fazekas score 2–3) was correlated with the risk of recurrent stroke, ischemic stroke, and hemorrhagic stroke. |

**Table 1.** *Cont.*

| Authors, Year of Publication | Type of Study | Biomarker | Type of Stroke | Number of Participants/Age of Participants (Years)/Gender (M/F) | Cerebrovascular Risk Factors (*n*) | Medication (*n*) | Previous Stroke (*n*) | Follow-Up Time | Leukoaraiosis /WMH Assessment | Severity of Leukoaraiosis | Time of MRI Execution/Specification of MRI | Scale of Stroke Severity and Prognosis/Clinical Outcome | Main Results |
|---|---|---|---|---|---|---|---|---|---|---|---|---|---|
| Chen et al., 2021 [40] | Longitudinal (prospective) | WMH, lacunes, CMBS, PVS | AIS (NIHSS ≤ 3) | 388/66.54 ± 11.15 151M/237F | Hypertension (324) Coronary artery disease (51) Atrial fibrillation (52) Diabetes mellitus (142) Hyperlipidemia (162) Hematencephalon (40) Smoking (73) Alcohol (56) Carotid atherosclerosis (259) | n/a | 61 | 90 days | Fazekas scale | n/a | Within 7 days of stroke onset/3.0 T MRI scanner with an 8-channel receiver array head coil. | NIHSS on admission, mRS at 90 days | Higher Fazekas scores were not correlated with stroke progression or stroke recurrence. |
| Nam et al., 2017 [41] | Longitudinal (prospective) | WMH, CMBs | AIS | 959/66 ± 12 590M/369F | Hypertension (630) Hyperlipidemia (295) Diabetes mellitus (291) Atrial fibrillation (138) Smoking (321) Alcohol (382) Family history of stroke (185) | Antiplatelets (864) Anticoagulants (121) Anti-hypertensives (470) Statin (680) Carotid endarterectomy or carotid stenting (13) | - | 2 years | Fazekas scale | severe | Within 24 h of admission/1.5- or 3.0- Tesla magnetic resonance scanners (Achieva 1.5T and 3.0T, Philips, Amsterdam, The Netherlands) | NIHSS | The recurrence group was more likely to have severe WMH and this association remained significant after adjusting confounders. |
| Ryu et al., 2017 [42] | Longitudinal (prospective) | Leukoaraiosis | AIS | 5035/66.3 ± 12.8 3000M/2035F | Hypertension (2474) Diabetes (1083) Hyperlipidemia (1284) Smoking (1665) Coronary artery disease (333) Atrial fibrillation (762) | Statins (440) Antiplatelets (790) | - | 3 months | Volumetric | Advanced/mild to moderate | n/a/1.5 T (*n* = 4327) or 3.0 T (*n* = 708) MRI systems | NIHSS on admission, mRS pre-stroke, at discharge and at 3 months. | Regardless of stroke subtype, WMHs were not associated with stroke recurrence within 3 months of follow-up. |

**Table 1.** *Cont.*

| Authors, Year of Publication | Type of Study | Biomarker | Type of Stroke | Number of Participants/Age of Participants (Years)/Gender (M/F) | Cerebrovascular Risk Factors (*n*) | Medication (*n*) | Previous Stroke (*n*) | Follow-Up Time | Leukoaraiosis /WMH Assessment | Severity of Leukoaraiosis | Time of MRI Execution/Specification of MRI | Scale of Stroke Severity and Prognosis/Clinical Outcome | Main Results |
|---|---|---|---|---|---|---|---|---|---|---|---|---|---|
| | | | | | Patients with minor cerebrovascular events | | | | | | | | |
| Ren et al., 2018 [43] | Longitudinal (prospective) | WMH | TIA | 181/61.4 ± 11.1 108M/73F | Hypertension (97)) Diabetes mellitus (41) Hyperlipidemia (63) Smoking (45) Coronary artery disease (12) Atrial fibrillation (7) | Antithrombotics (96) Statins (97) | - | 1 year | Volumetric | Mild to severe | Within 2 days of symptom onset/1.5 Tesla scanner (Philips Intera Master Medical Systems, Best, The Netherlands) | - | The presence of baseline WMLs increased recurrent vascular incidences (recurrent TIA, stroke, myocardial infarction, and vascular death) by up to 4.2-fold and the correlation between WML volumes and recurrent risks persisted after controlling for age, hyperlipidemia, diabetes mellitus, duration of symptoms, and anti-thrombotics. |
| Lim et al., 2015 [44] | Longitudinal (prospective) | CMBS | TIA | 500/64 291M/209F | Hypertension (333) Diabetes mellitus (149) Hyperlipidemia (157) Atrial fibrillation (53) Coronary artery disease (42) Smoking (131) Family history of stroke (104) | Antiplatelets (493) Anticoagulation (110) Statins (345) | 87 | 90 days | Fazekas scale | n/a | Within 48 h after symptom onset/n/a | - | Among MRI variables, WMHs and CMBs were significantly associated with a higher risk of future ischemic stroke. |

**Table 1.** *Cont.*

| Authors, Year of Publication | Type of Study | Biomarker | Type of Stroke | Number of Participants/Age of Participants (Years)/Gender (M/F) | Cerebrovascular Risk Factors (*n*) | Medication (*n*) | Previous Stroke (*n*) | Follow-Up Time | Leukoaraiosis /WMH Assessment | Severity of Leukoaraiosis | Time of MRI Execution/Specification of MRI | Scale of Stroke Severity and Prognosis/Clinical Outcome | Main Results |
|---|---|---|---|---|---|---|---|---|---|---|---|---|---|
| Foschi et al., 2020 [45] | Longitudinal (prospective) | Leukoaraiosis | TIA | **Single TIA: 822/**71.1± 14.8 431M/391F **Early recurrent TIA: 231/**69.0 ± 13.5 130M/101F | Hypertension (710) Dyslipidemia (436) Diabetes mellitus (166) Acute/Chronic heart failure (31) Smoking (134) Alcohol (31) Coronary disease (36) Aortic Arch pathology (14) Extracranial ICA stenosis (173) Peripheral artery disease (47) Atrial fibrillation (137) | Single antiplatelet (437) Dual antiplatelet (13) Anticoagulants (104) Anti-hypertensives (672) Lipid-lowering (286) | 237 | 60 months | van Swieten scale | n/a | n/a | - | Independent predictors of stroke at 3 and 12 months in the early recurrent TIA subgroup included the presence of TIA with an acute ischemic lesion, clinical presentation with dysarthria, and leukoaraiosis. |
| Wardlaw et al., 2017 [46] | Longitudinal (prospective) | WMH | AIS (with NIHSS ≤7) | 190/65.3 ± 11.3 112M/78F | Diabetes mellitus (21) Hyperlipidemia (116) Smoking (73) | n/a | n/a | 1 year | Fazekas scale | n/a | At presentation and at 1 year after stroke/1.5T Signa HDxt, General Electric, Milwaukee, WI) with self-shielding gradients and an 8-channel phased-array head coil. | NIHSS on admission, mRS at 1 year | Any recurrent stroke or TIA was associated with WMH growth rather than reduction. |
| Zerna et al., 2018 [47] | Longitudinal (prospective) | WMH | TIA or AIS (NHISS ≤3) | 412/67.3 248M/164F | Congestive heart failure (3) Atrial fibrillation (28) Diabetes mellitus (57) Current smoking (63) Past smoking (91) | Aspirin (134) Plavix (24) Aggrenox (4) Warfarin (17) | n/a | 90 days | Fazekas scale, volumetric | n/a | n/a/GE 3-T scanner or a Siemens 1.5-T MR scanner. | NIHSS on admission, mRS at 90 days | Neither higher Fazekas scale scores nor higher WMH volume were associated with stroke progression, TIA recurrence, or stroke recurrence. |

**Table 1.** *Cont.*

| Authors, Year of Publication | Type of Study | Biomarker | Type of Stroke | Number of Participants/Age of Participants (Years)/Gender (M/F) | Cerebrovascular Risk Factors (*n*) | Medication (*n*) | Previous Stroke (*n*) | Follow-Up Time | Leukoaraiosis /WMH Assessment | Severity of Leukoaraiosis | Time of MRI Execution/Specification of MRI | Scale of Stroke Severity and Prognosis/Clinical Outcome | Main Results |
|---|---|---|---|---|---|---|---|---|---|---|---|---|---|
| | | | | | **Patients with Intracerebral Hemorrhage (IH)** | | | | | | | | |
| Park et al., 2021 [48] | Longitudinal (prospective) | WMH | IH | 1454/65.9 ±10.8 904M/550F | Hypertension (1294) Dyslipidemia (626) Diabetes mellitus (465) Coronary heart disease (67) Smoking (644) Family history of stroke (314) | Antihypertensive (1069) Statin (1160) Probucol (732) Cilostazol (719) Aspirin (735) | n/a | 1.9 years | Fazekas scale | Mild-moderate/advanced | At baseline and after the 13-month visit/n/a | NIHSS at baseline | Compared with the reference group, patients with prior ICH and advanced WMH had an over 2.5-fold increased risk of recurrent ischemic stroke and an over 30-fold risk of hemorrhagic stroke during the 2-year follow-up period. |
| | | | | | **Patients with Any Stroke Subtype** | | | | | | | | |
| Kumral et al., 2015 [49] | Longitudinal (prospective) | Leukoaraiosis | Any type | 9522/65 ± 12 5387M/4135F | Hypertension (7388) Smoking (1678) Diabetes mellitus (2810) Obesity (1235) Coronary heart disease (1192) Atrial fibrillation (1645) Hyperhomocysteinemia (1175) Hyperuricemia (729) Hypercholesterolemia (3481) Hypertriglyceridemia (2240) Higher LDL-cholesterol (4339) Lower HDL-cholesterol (5755) Lower apo A (274) Higher apo B (1364) Higher lipoprotein (931) | n/a | n/a | 5 years | Fazekas scale | Mild/moderate/severe | n/a | NIHSS at admission, mRS at discharge | Stroke recurrence was significantly associated with severe LA, especially for patients in the large-artery disease, small-artery disease or intracerebral hemorrhage subgroups. |

**Table 1.** *Cont.*

| Authors, Year of Publication | Type of Study | Biomarker | Type of Stroke | Number of Participants/Age of Participants (Years)/Gender (M/F) | Cerebrovascular Risk Factors (*n*) | Medication (*n*) | Previous Stroke (*n*) | Follow-Up Time | Leukoaraiosis /WMH Assessment | Severity of Leukoaraiosis | Time of MRI Execution/Specification of MRI | Scale of Stroke Severity and Prognosis/Clinical Outcome | Main Results |
|---|---|---|---|---|---|---|---|---|---|---|---|---|---|
| Imaizumi et al., 2014 [50] | Longitudinal (prospective) | WML | Any type | 807/69.8 ± 12.0 456M/351F | Hypertension (511) Hemodialysis (32) Diabetes mellitus (185) | Antiplatelets (426) Warfarin (105) | 224 | 31.6 ± 22.2 months | Fazekas scale | Grade 0–3 | n/a | - | The incidence of stroke recurrence presenting as ICHs, lacunar infarctions, and atherothrombotic infarctions, but not as cardioembolic infarctions, was significantly higher in patients with high-grade WML. |
| Imaizumi et al., 2015 [51] | Longitudinal (prospective) | WML | Lacunar infarctions | 305/70.2 6 ± 11.7 167M/138F | Hypertension (189) Hemodialysis (8) Diabetes mellitus (94) Smoking (90) | Antiplatelets (287) Warfarin (17) Statin (66) | - | 50.7 ± 32.8 months | Fazekas scale | Grade 0–3 | n/a | - | The incidences of stroke recurrence presenting as lacunar infarctions or deep ICHs were significantly higher in patients with high-grade WMLs, but there was no such relation for recurrences presenting as lobar ICHs, atherothrombotic infarctions, or cardioembolic infarctions. |
| **Patients with Atrial Fibrillation** | | | | | | | | | | | | | |
| Du et al., 2021 [52] | Longitudinal (prospective) | WML | TIA or AIS | 1419/75.8 ± 10.4 822M/597F | Hypertension (885) Hyperlipidemia (622) Diabetes mellitus (237) Chronic heart failure (59) Smoking (699) Alcohol (968) Ischemic heart disease (233) Vascular disease (265) Atrial fibrillation (602) | Anticoagulation (1373) | 292 | 24 months | Fazekas scale | Moderate to severe | At baseline/n/a | - | The presence of moderate-to-severe WMH was not significantly associated with ischemic stroke during follow-up. |

**Table 1.** *Cont.*

| Authors, Year of Publication | Type of Study | Biomarker | Type of Stroke | Number of Participants/Age of Participants (Years)/Gender (M/F) | Cerebrovascular Risk Factors (*n*) | Medication (*n*) | Previous Stroke (*n*) | Follow-Up Time | Leukoaraiosis /WMH Assessment | Severity of Leukoaraiosis | Time of MRI Execution/Specification of MRI | Scale of Stroke Severity and Prognosis/Clinical Outcome | Main Results |
|---|---|---|---|---|---|---|---|---|---|---|---|---|---|
| Hert et al., 2020 [53] | Longitudinal (prospective) | WMH, CMBs | IS | 320/78.2 ± 9.2 170M/150F | Hypertension (241) Hypercholesterolemia (122) Diabetes mellitus (62) Smoking (81) Alcohol (78) | DOAC (216) DOAC/antiplatelet (18) VKA (61) VKA/antiplatelet (15) Antiplatelets (5) | n/a | Median follow-up time of 754 days | ARWMC score | n/a | n/a | NIHSS at baseline, mRS at 3, 6, 12, and 24 months | WMHs were associated with an increased risk for recurrent ischemic stroke and intracranial hemorrhage. |
| Kashima et al., 2018 [54] | Longitudinal (prospective) | WMH | ESUS | 236/70.2 ± 12.1 141M/95F | Hypertension (178) Dyslipidemia (95) Diabetes mellitus (68) Chronic heart failure (11) Smoking (97) Dialysis (7) | Antiplatelet therapy (145) Anticoagulant therapy (warfarin) (81) | 27 | From 7 days to 12.9 years (median 54.3 months) | DSWMH classification developed by Shinohara et al. [55] | n/a | At the time of first admission and at recurrence/1.5-Tscan-ners | NIHSS at admission | DSWMH grade greater than or equal to 3 was an independent predictor of recurrent ischemic stroke. |

### *3.3. Method of Leukoaraiosis Neuroimaging Assessment*

In total, thirteen studies preferred the Fazekas scale score, one study used the van Swieten scale, one study applied the age-related white matter change scale, one study used the deep and subcortical white matter hyperintensities (DSWMH) classification developed by Shinohara et al. [55], and four studies estimated the WMH volume on MRI.

### *3.4. Study Design*

In total, all the studies included in this review were longitudinal. They were either retrospective or prospective cohorts.

### *3.5. Stroke Patient Groups and Demographic Profile*

The total number of stroke patients included in all studies ranged from $n = 181$ [45] to $n = 9522$ [51]. Across the 19 studies, eight studies had a disease sample size between 100 and 500 patients, three studies between 501 and 1000, five studies between 1001 and 5000, and three studies had a disease sample size larger than 5000 patients. The mean/median patient age ranged from 59.5 [36] years to 78.2 [53] years.

### *3.6. Reference Groups*

None of the 37 included studies involved stroke patients being contrasted to demographically matched healthy individuals and none of the studies included a disease-control group other than stroke patients.

### *3.7. Scales of Stroke Severity and Prognosis/Clinical Outcome*

The National Institutes of Health Stroke Scale (NIHSS) and modified ranking scale (mRS) were used simultaneously in five studies; the NIHSS alone was used in six studies. Moreover, the NIHSS, the Trial of Org 10172 in Acute Stroke Treatment (TOAST), and the Oxfordshire Community Stroke Project (OCSP) classifications were used in one study. Lastly, the NIHSS and the mini-mental state examination (MMSE) were utilized in one study.

## 4. Discussion

A systematic review of research over the last decade was conducted to explore the value of LA as a biomarker for stroke recurrence. Nineteen complete-text original studies addressing the potential value of monitoring LA volume for stroke recurrence have been identified and categorized according to stroke subtype and whether patients had atrial fibrillation.

### *4.1. Stroke Recurrence in Patients with Ischemic Stroke (IS)*

The main finding of Andersen et al. [36] was that increasing WMH volume is associated with the risk of recurrence of IS in patients without AF. In particular, increasing periventricular and DWMH burden and Fazekas total score were associated with recurrence. The results contrast with a previous study from the Athens Stroke Registry, in which the addition of LA to known clinical scores did not increase their accuracy. According to the authors, this is most likely due to differences in the study populations and thresholds established to indicate the presence of WMH. Another interesting finding is that they observed no advantage of the Fazekas total score over the DWMH score, with the simplicity of the DWMH score being an advantage. A DWMH score equal to or greater than 2 stands for moderate-to-severe changes and represents approximately 25% of patients. The correlation was valid even after adjustment for known stroke risk factors, and thus they believe that WMH plays an independent role, expressing individual cerebral vulnerability to the overall burden of vascular risk factors. The inclusion of WMH in known clinical scores is thought to improve their predictability in stroke patients, although risk factors for leukoaraiosis are already included in these scores.

Similarly, Kim et al. [37] proposed that after controlling for the severity of the initial stroke, standard stroke risk factors, underlying stroke mechanism, and preventive stroke medication, patients with extensive LA were 1.5 times more likely to experience another IS in the short term than those without. These results can be explained by the fact that asymptomatic new infarcts are much more frequent in the first days after stroke than symptomatic ones. Severe LA might make it easier for asymptomatic infarcts to become symptomatic because of the impaired brain's ability to tolerate ischemia and its lower capacity to make up for lost function. In support of this assumption, they reported that patients with pathogenesis at high risk for asymptomatic recurrence were two to three times more likely to have a symptomatic recurrence if significant LA was present. Regarding the location of LA, they found that periventricular LA was associated with a slightly increased risk of early recurrence compared with subcortical LA. They believe that these may be two pathophysiologically distinct types of WM injury. Periventricular LA relates to vasomotor reactivity and subsequent cerebral hypoperfusion, and subcortical LA is associated with microangiopathy.

On the same basis, Ryu et al. [38] found that increasing the volume burden of white matter hyperintensities was associated with recurrent stroke after IS within the 1-year follow-up period. Specifically, in agreement with another study, the 1-year recurrence rate was 6.7 %/year, and the association was stronger for recurrent HS than for IS. Notably, the HS risk in the lowest quartile of WMH was approximately 18 times higher than the risk in the highest quartile. Considering this, the authors emphasize that this finding is in stark contrast to the observation that patients with increased WMH volume are more likely to receive more antithrombotic therapy at discharge. This suggests that WMH burden may play a role in determining the most suitable therapy plan for patients after stroke and in personalizing the risk–benefit ratio.

Furthermore, Park et al.'s [39] examined WMH and the risk of having another stroke in patients with ischemic disease of the small vessels. Consistent with previous studies, they concluded that moderate-to-severe WMH was linked with a more than twofold increased risk of recurrent stroke over the follow-up phase, in contrast to the group with less-severe WMH. Furthermore, an independent connection between advanced WMH and both ischemic and hemorrhagic stroke was found when stratified by stroke subtype. Although the mechanism is not clear, they hypothesize that their findings are caused by decreased WM blood flow and vascular reserve, as well as the fragility of the subcortical arteriolar bed. In addition, they hypothesize that the association may be influenced by the more disabling effects of index stroke and restricted mobility due to the impaired microstructure of the WMH. Moreover, baseline WMH was strongly connected to an elevated rate of dementia, which is considered a potential marker of future stroke risk, and advanced WMH may have resulted in insufficient medication adherence, leaving patients exposed to repeated stroke.

On the same note, Chen et al. [40] attempted to explain the association between neuroimaging biomarkers of small-vessel disease of the brain, such as LA, and stroke patients' short-term results. In contrast to the previous study, their results showed that a greater Fazekas Scale score at baseline was connected with worse functional outcomes but not with stroke progression and recurrence. On the same basis, Ryu et al. [42] studied stroke outcomes in patients with large LA volumes and found no correlation between WMH and stroke recurrence within three months of the first stroke's commencement, regardless of stroke subtype.

Apart from that, Nam et al. [41] found that instead of the usual cardiovascular risk factors, abnormal neuroimaging findings, including severe WMH, were associated with long-term stroke recurrence. However, they did not find statistical relevance for short-term recurrence. These findings suggest that scores from LA may identify patients at increased risk who require additional monitoring and comprehensive medical treatment.

*4.2. Stroke Recurrence in Patients with Minor Cerebrovascular Events*

In their study, Ren et al. [43] investigated the predictability of WML in patients with TIA. They first found that WMH volumes were higher in TIA patients compared with the control group. Their primary finding was that baseline WML increased the number of recurrent vascular events by up to more than fourfold, and that this increase persisted with increasing WML volume, reaching an 8.5-times increase in the last quartile compared with the first quartile. The authors believe that this may be the reason why, even after a measurable reduction in stroke risk, the rate of recurrent vascular events in patients with WMLs remains unacceptably high. In conclusion, they believe that WMLs can serve as a radical biomarker for TIA recurrence and suggest that although there is no definitive effective treatment, good blood pressure control, antiplatelet agents, and endothelial stabilization may be feasible.

On the same basis, Lim et al. [44] explored the stroke recurrence rate of TIA patients, 90 days post-stroke onset, and the importance of prognostic factors like WMH and CMB. The authors reported a significant association between the imaging biomarkers (WMHs, CMBs) and stroke recurrence, although the recurrent stroke rate was 5%, which was lower than the median rate of similar previous studies. This may be because WMH and CMB share risk factors for a later ischemic stroke with other diseases of the small vessels and are considered to result from general microvascular instabilities and endothelial fragility of the brain. Additionally, Foschi et al. [45] evaluated the early recurrence of TIA by examining characteristics such as frequency, clinical, demographic, and etiologic factors. In particular, in individuals with early recurrent TIA, extensive leukoaraiosis predicted stroke independently, both at 3 months and at 12 months, especially in patients without atrial fibrillation.

Moreover, Wardlaw et al. [46] demonstrated that lower WMH values in patients with IS were linked with a lower number of recurrent strokes and other vascular incidents at one year, with parallel changes in various tissue characteristics when compared to WMH-growth participants. In addition, they provide evidence that preventing the exacerbation of brain damage due to white matter hyperintensities may lead to long-term brain-health benefits, thus the better management of risk factors could not only attenuate WMH growth but also reverse some WMH-related brain damage. The authors believe that lipohyalinosis, arteriolosclerosis, and fibrinoid necrosis are characterized by perforating arteriolar wall destruction, luminal constriction, and occlusion, all of which can result in persistent reduced brain blood flow and ischemia, and the disruption of the blood–brain barrier can result in perivascular edema and consequent damage to the brain, leading to LA. According to their results, they suggest that the reversibility of WMH, not just the lack of growth, has major implications for the development of methods to prevent long-term brain damage associated with stroke. To uncover the processes underlying WMH dynamics, future investigators should examine microstructure in areas of WMH development using precise cerebral and systemic vascular function tests. Similarly, Zerna et al.'s [47] study regarding WMH in patients with minor cerebrovascular events found that greater Fazekas scale scores, larger WML values, and a greater WML volume ratio at baseline were not linked with stroke progression, TIA recurrence, or stroke recurrence.

*4.3. Stroke Recurrence in Patients with Intracerebral Hemorrhage (ICH)*

With respect to LA, as a biomarker, Park et al. [48] investigated the association between WMH and previous ICH in relation to the risk of recurrent stroke in non-cardioembolic patients, because both covariates are known to be linked with ischemic stroke. One of the main findings of the study was that patients with prior ICH and advanced WMH had a greater than 2.5-fold higher risk of major cardiovascular events and ischemic stroke recurrence than patients in the reference group with mild/moderate LA. The risk of recurrent stroke was likewise increased in the first group compared to the group of individuals with advanced WMH but no ICH. Another interesting finding of the investigators was that patients with advanced WMH had a risk of recurrent hemorrhagic stroke more than 30-times higher than

patients with moderate WMH. This could be due to older age, higher hypertension, and greater WMH and CMB burden in the ICH/advanced WMH patients. They considered these results as evidence of an additional relationship between ICH and WMHs.

### 4.4. Stroke Recurrence in Any Stroke Subtype

Regarding the association between LA and stroke recurrence, the most important finding of the research conducted by Kumral E. et al. [49] was that the stroke recurrence rate after one year for all the subtypes was much higher in those with leukoaraiosis than in those without. The same was true 5 years after stroke onset for all subtypes, except stroke of cardioembolic origin and other rare subtypes. The association between small-vessel disease infarcts and LA may be explained by the fact that LA areas have impaired brain perfusion and vascular resistance, which may lead to recurrent small-vessel disease-related events in the periventricular or subcortical regions. The recurrent stroke was of the same subtype in most cases; in 20% of patients with intracerebral hemorrhage (ICH), the recurrent stroke subtype was an IS, most frequently a small-vessel disease in patients with LA. This strong association between ICH and small-vessel disease with LA suggests that they may have shared risk factors, and along with the increased ICH events due to tissue plasminogen activator in LA patients, this may be the cause of future stroke recurrences. Similarly, Imaizumi et al. [50] examined the incidence of all stroke recurrences, ICH, lacunar infarcts, and atherothrombotic infarcts. To quantify LA, they used the Fazekas score grades 1–3, and patients with grades 2 and 3 had significantly higher rates than patients with a lower Fazekas score. Thus, they concluded that the presence of high-grade WMLs increases the rate of stroke recurrence, which presents as cerebral hemorrhage, lacunar, and atherothrombotic infarcts but not infarcts due to cardioembolism. They based their findings on the fact that WMLs can be caused by both amyloid angiopathy and lobar brain hemorrhage, as well as hypertensive microangiopathy with deep ICHs and lacunar infarcts. In another similar study, Imaizumi et al. [51] specifically examined patients with a history of lacunar infarcts and the role of WMLs on the type of recurrent stroke. Again, the presence of high-grade WMLs was found to increase the rate of stroke recurrences presenting as lacunar infarcts and deep ICHs, but this time no other types of stroke recurrence were statistically significant. Although a reduction in cerebral blood flow due to middle cerebral artery stenosis has been associated with WML and WML may be a predictive factor for the severity of cerebral macroangiopathy, it did not increase the incidence of atherothrombotic infarcts in this study. The significance of these studies lies in the potential ability of WML values to predict the occurrence of ICHs and lacunar infarctions related to microangiopathies; however, these studies included only the Asian population, and therefore they may only be applicable in Asia as some aspects of stroke and risk factors may differ between regions.

### 4.5. Stroke Recurrence in Patients with Atrial Fibrillation

Regarding the use of markers for small-vessel disease (WMH, BGPVS, CMB), Du et al. [52] investigated the risk of recurrent ischemic stroke during follow-up in patients receiving anticoagulant treatment for AF. The investigators report that they found a weak indication that the existence of WMH increases the rate of ischemic stroke during follow-up. Similarly, Hert et al. [53] found that WMH was associated with an elevated risk for a combined outcome, which is defined as recurring IS, ICH, or death. In addition, the total risk for reoccurring ischemic stroke was greater than for ICH. Both studies suggest that the association between LA and anticoagulant therapy in relation to stroke recurrence should be thoroughly investigated in the future, because, although Du et al. [52] found no significant association, Hert et al. [53] reported that the presence of WMH was associated with higher stroke risk in patients treated with anticoagulants.

On the same note, Kashima et al. [54] investigated the connection between WMH and stroke recurrence and new-onset AF in patients who had suffered an embolic stroke of an undetermined source. The recurrent stroke was also an embolic stroke of an undetermined source in most cases. On this basis, they demonstrated a correlation between embolic stroke of unknown origin and WMH and proposed that WMH might be utilized as a surrogate

marker for recurrent ischemic stroke and to predict hidden AF in patients with embolic stroke of unknown origin.

## 5. Limitations

Our systematic review is not without limitations. First, most studies included only patients with mild-to-moderate stroke, which restricts the results' adaptability to the overall stroke populace. Additionally, those with significant impairments or those who were deemed unfit for an MRI were frequently ruled out, as were those suffering from dementia who were incapable of giving their permission. As a result, we offered a comprehensive analysis of patients' sociodemographic and clinical data, which gave an estimation of their impact on acute stroke and provided a foundation for additional research. We acknowledge, however, that most study populations are from Asia and their results may not be generalizable to the world population. Although we made every effort to avoid personal bias, we cannot exclude the possibility that some degree of bias may be present in our review.

## 6. Conclusions

Considering all aspects, the present review provides a comprehensive overview of the possible clinical applications of LA as a biomarker in stroke recurrence. Our 17 findings support the utility of LA volume measurement in patients who have suffered an acute stroke and show that a biomarker-based strategy based on LA might provide valuable insights regarding recurrent strokes and significantly alter individualized stroke treatment, particularly in patients with anticoagulants. LA appears to serve as a surrogate marker not only for WM blood flow and vascular reserve but also for stroke recurrence and, therefore, can reliably assess the likelihood of acute stroke recurrence and the patient's specific stroke subtype. It was found that elevated WMH-WML values are usually associated with a higher risk of stroke recurrence than in patients with lower volumes. It is essential to keep in mind that the figures shown above are not confined to those with acute ischemic stroke and that equivalent outcomes have been found in patients with TIA and HS-ICH. Given that assessing LA provides crucial predictive information beyond clinical features, it could greatly improve our understanding of stroke patients, help them and their families with counseling, and improve the proper diagnosis of stroke by strengthening current clinical scores with the addition of WMH volumes. It could also lead to the effective selection of more suitable radical therapy with specific brain-protective drugs, and thus to better stroke care overall. Further studies on the association between LA and stroke recurrence are needed to better understand this clinically significant relation. In particular, the connection between radiological biomarkers such as LA and long-term outcomes in individuals with mild cerebrovascular episodes has not yet been adequately demonstrated, and more detailed longitudinal studies in large populations should be performed. Moreover, with the use of new neuroimaging techniques, researchers should further evaluate the relationship between LA and the recurrence of specific stroke subtypes.

**Author Contributions:** Conceptualization, D.T. and F.C.; methodology, T.D.; validation, E.M.; formal analysis, S.R.; investigation, M.A.; resources, A.S.; data curation, I.O.; writing—original draft preparation, E.P.; writing—review and editing, M.K., I.K. and C.N.; visualization, C.K.; supervision, A.G. and E.A.P.; project administration, S.K.; funding acquisition, N.A. and K.V. All authors have read and agreed to the published version of the manuscript.

**Funding:** We acknowledge the support of this work by the project "Study of the interrelationships between neuroimaging, neurophysiological and biomechanical biomarkers in stroke rehabilitation (NEURO-BIO-MECH in stroke rehab)" (MIS 5047286), which was implemented under the action "Support for Regional Excellence" funded by the operational program "Competitiveness, Entrepreneurship and Innovation" (NSRFm2014-2020) and co-financed by Greece and the European Union (the European Regional Development Fund).

**Institutional Review Board Statement:** Not applicable.

**Informed Consent Statement:** Not applicable.

**Data Availability Statement:** All data discussed within this manuscript are available on PubMed.

**Conflicts of Interest:** The authors declare no conflict of interest.

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
