# Peer review of "Leukoaraiosis as a Promising Biomarker of Stroke Recurrence among Stroke Survivors: A Systematic Review"

_2035-8377, doi:10.3390/neurolint15030064_

Round 1

Reviewer 1 Report

Dear Authors, 

I have had the opportunity to review your manuscript titled "Leukoaraiosis as a Biomarker for Stroke Recurrence: A Systematic Review," and I must commend you on the thoroughness and importance of the study. The topic is of great significance in the field of stroke research, and your findings offer valuable insights into the potential utility of leukoaraiosis as a prognostic biomarker.

While the manuscript is well-written and the research is commendable, I have a few suggestions that, in my opinion, could further strengthen the article and enhance its impact. These suggestions aim to provide more context, clarity, and transparency, and I believe they will significantly contribute to the overall quality of the work.

1. Abstract: - Consider providing specific details, such as the total number of patients included in the review and the main findings for each stroke subtype studied, to offer a more comprehensive overview.

Add a sentence summarizing the clinical implications of the findings, emphasizing the potential impact of using leukoaraiosis as a biomarker for stroke recurrence in clinical practice.

2. Introduction:

   - Include more recent statistics and data on the global burden of stroke and its impact on disability and mortality to provide a clearer context for the study's importance.

   - Expand the section on biomarkers, offering a more comprehensive overview of different types of biomarkers studied in stroke research and their potential utility in predicting stroke recurrence.

3. Materials and Methods:

   - Provide more specific details on the search strategy, such as the specific search terms used, inclusion/exclusion criteria, and any additional filters or limits applied during the database search.

   - Explain the reasons for choosing the specific databases (MEDLINE and Scopus) and the time frame for the search (January 1, 2012, to June 25, 2022).

   - Add more information on the data extraction process, including the process of data verification and potential measures taken to address any disagreements during the screening and selection process.

4. Results:

   - Provide more specific details on individual study characteristics, such as study design, patient demographics, and follow-up duration, to enhance the reader's understanding of the included studies.

   - Consider including a table summarizing the main characteristics of each included study for quick reference.

5. Discussion:

   - Strengthen the discussion by explicitly connecting the findings to the results presented in the results section.

   - Highlight the clinical implications of the findings and potential applications of using leukoaraiosis as a prognostic biomarker for stroke recurrence.

   - Integrate the limitations of the study throughout the discussion, specifically when discussing the relevance and generalizability of the findings.

6. Conclusion:

   - Restate the key clinical implications of using leukoaraiosis as a biomarker for stroke recurrence in a more concise manner.

   - Emphasize the importance of further research in this area to advance our understanding of leukoaraiosis as a prognostic tool for stroke survivors.

I believe that implementing these suggestions will significantly improve the manuscript's overall quality and impact. Your research has the potential to contribute greatly to the field of stroke research, and I commend your efforts in conducting this systematic review.

Thank you for considering these recommendations.

Best regards,

The Reviewer

Author Response

Dear Reviewer 1:

Many thanks for your prompt response and your time spent reviewing our manuscript.

Your comments were thoroughly investigated and appropriate modifications were made to the text as follows:

  1. Abstract: - Consider providing specific details, such as the total number of patients included in the review and the main findings for each stroke subtype studied, to offer a more comprehensive overview.

According to your suggestions, this was added and is showcased in blue

“data collected from a total of 34,546 stroke patients. A higher rate of extended LA was strongly associated with stroke recurrence in all stroke subtypes, even after adjustment for clinical risk factors. In particular, patients with ischemic stroke or transient ischemic attack with advanced LA had a significantly higher risk of future ischemic stroke, whereas patients with previous intracerebral hemorrhage and severe LA had a more than 2.5-fold increased risk of recurrent ischemic stroke and a more than 30-fold increased risk of hemorrhagic stroke. Finally, in patients receiving anticoagulant treatment for AF, the presence of LA was associated with an increased risk of recurrent ischemic stroke and intracranial hemorrhage. Because of this valuable predictive information, evaluating LA could significantly expand our knowledge of stroke patients and, thereby improve overall stroke care.”

  1. Add a sentence summarizing the clinical implications of the findings, emphasizing the potential impact of using leukoaraiosis as a biomarker for stroke recurrence in clinical practice.

According to your suggestions, this was added and is showcased in blue

“Because of this valuable predictive information, evaluating LA could significantly expand our knowledge of stroke patients and, thereby improve overall stroke care”

  1. Introduction
  • - Include more recent statistics and data on the global burden of stroke and its impact on disability and mortality to provide a clearer context for the study's importance.

According to your suggestions, more recent statistics were added and are showcased in blue.

“Moreover, a recent systematic analysis considering data from the last three decades shows that the burden of stroke is steadily increasing, as the absolute number of cases has increased by 70% compared to 1990 [3]”

  1.  - Expand the section on biomarkers, offering a more comprehensive overview of different types of biomarkers studied in stroke research and their potential utility in predicting stroke recurrence.

According to your suggestions, this was added and is showcased in blue

“Others may point to systemic inflammation which increases the rate of stroke recurrence through various mechanisms such as inducing a hypercoagulable state or via causing endothelial damage. This category includes Interleukin-6 (IL-6) and YKL-40 which were demonstrated in a separate study to be linked with stroke recurrence [21]. Another promising biomarker for stroke recurrence appears to be the amount of lipoprotein A detected in the blood of the patient, a correlation that is however attenuated when LDL-c and inflammation markers remain low [22]. There are also neuroimaging biomarkers that provide an indirect look at the brain reserve via examining the lesions that are present in case of deterioration and loss of healthy brain tissue. Such biomarkers are cerebral microbleeds which have been correlated with stroke recurrence in a different study [23]”

  1. Materials and Methods:

   - Provide more specific details on the search strategy, such as the specific search terms used, inclusion/exclusion criteria, and any additional filters or limits applied during the database search.    

Requested Info is highlighted with blue

   - Explain the reasons for choosing the specific databases (MEDLINE and Scopus) and the time frame for the search (January 1, 2012, to June 25, 2022).

MEDLINE and Scopus are considered to be two of the most reliable and diverse databases in the medical field and thus they were chosen as ideal.

The timeframe that took only the last decade into consideration was put in place so the research is the most up-to-date it can be.

   - Add more information on the data extraction process, including the process of data verification and potential measures taken to address any disagreements during the screening and selection process.

According to your suggestion, this was done. The requested Info is highlighted with blue

  1. Results:

   - Provide more specific details on individual study characteristics, such as study design, patient demographics, and follow-up duration, to enhance the reader's understanding of the included studies.  

According to your suggestion, this was done. The requested Info is highlighted with blue

   - Consider including a table summarizing the main characteristics of each included study for quick reference.

According to your suggestion, this was done. The requested Info is highlighted with blue

  1. Discussion:

   - Strengthen the discussion by explicitly connecting the findings to the results presented in the results section.

According to your suggestion changes were made and are highlighted with blue:

The correlation was valid even after adjustment for known stroke risk factors, thus they believe that WMH plays an independent role, expressing individual cerebral vulnerability to the overall burden of vascular risk factors

This may be because WMH and CMB share risk factors for a later ischemic stroke with other diseases of the small vessels and are considered to result from general microvascular instabilities and endothelial fragility of the brain.

 severe WMH, were associated with long-term stroke recurrence. However, they didn't find statistical relevance for short-term recurrence. These findings suggest that scores from LA may identify patients at increased risk who require additional monitoring and comprehensive medical treatment

and fibrinoid necrosis is characterized by perforating arteriolar wall destruction, luminal constriction, and occlusion, all of which can result in persistent reduced brain blood flow and ischemia; disruption of the blood-brain barrier can result in perivascular edema and consequent damage to the brain

The risk of recurrent stroke was likewise increased in the first group compared to the group of individuals with advanced WMH but no ICH

   - Highlight the clinical implications of the findings and potential applications of using leukoaraiosis as a prognostic biomarker for stroke recurrence.

According to your suggestion changes were made and are highlighted with blue:

  1. Inclusion of WMH in known clinical scores is thought to improve their predictability in stroke patients, although risk factors for leukoaraiosis are already included in these score.
  2. This suggests that WMH burden may play a role in determining the most suitable therapy plan for patients after stroke and in personalizing the risk-benefit ratio.
  3. Moreover, baseline WMH was strongly connected to an elevated rate of dementia, which is considered a potential marker of future stroke risk, and advanced WMH may have resulted in insufficient medication adherence, leaving patients exposed to a repeated stroke.
  4. These findings suggest that scores from LA may identify patients at increased risk who require additional monitoring and comprehensive medical treatment.

   - Integrate the limitations of the study throughout the discussion, specifically when discussing the relevance and generalizability of the findings.

According to your suggestion changes were made and are highlighted with blue:

however these studies included only Asian population and therefore they may only be applicable in Asia as some aspects of stroke and risk factors may differ between regions.

  1. Conclusion:

   - Restate the key clinical implications of using leukoaraiosis as a biomarker for stroke recurrence in a more concise manner.

According to your suggestion changes were made and are highlighted with blue:

it could greatly improve our understanding of stroke patients, help them and their families with counseling, and improve the proper diagnosis of stroke by strengthening current clinical scores with the addition of WMH volumes. It could also lead to effective selection of more suitable radical therapy with specific brain-protective drugs, and thus to better stroke care overall.

   - Emphasize the importance of further research in this area to advance our understanding of leukoaraiosis as a prognostic tool for stroke survivors.

According to your suggestion changes were made and are highlighted with blue:

Further studies on the association between LA in patients with stroke is needed to better understand this clinically significant relation. In particular, the connection between ra-diological biomarkers such as LA and long-term outcomes in individuals with mild cerebrovascular episodes has not yet been adequately demonstrated, and more detailed longitudinal studies in large populations should be performed. Moreover, with the use of new neuroimaging techniques, researchers should further evaluate the relationship between LA and the recurrence of specific stroke subtypes.

Thank you for the precise guidance and helpful suggestions.

Yours Sincerely,

Dr Tsiptsios

Reviewer 2 Report

The manuscript “Leukoiarosis as a promising biomarker of stroke recurrence among stroke survivors” is a systematic review.  I have only some minor comments, these are the following:

·       The Abstract is concise and summarizes the main aim of the study.

·       The Introduction is long, it should be shortened.

·       Methods:

-The authors followed and documented PRISMA checklist.

·       Results:

-The results are summarized in a table, and also detailed in the manuscript.

The severity pf LA is given as e.g. moderate, severe. It would be also useful if it would be given e.g. in the table what did the authors of publications mean by LA being mild/moderate/ severe (similar with Fazekas score)?

Probably the specification of the MRIs are given in the publications, was there a difference? At least it should be given in a sentence or even better in the table.

·       Discussion

-17 publications are mentioned, 19 were enrolled, probably typo.

-The summary is important, and relevant according to the review. Nevertheless, most of the studies are from Asia (13/19) /5 from Europe, 1 from America/. It is important to mention in the discussion part as well, since it is known that some aspects of stroke and risk factors may differ between these regions.

Author Response

Dear reviewer 2,

Many thanks for your prompt response and your time spent reviewing our manuscript.

Your comments were thoroughly investigated and appropriate modifications were made to the text as follows:

  • The Introduction is long, it should be shortened.

According to your suggestion, Intro was shortened

  • Methods:

-The authors followed and documented PRISMA checklist.

  • Results:

-The results are summarized in a table, and also detailed in the manuscript.

The severity pf LA is given as e.g. moderate, severe. It would be also useful if it would be given e.g. in the table what did the authors of publications mean by LA being mild/moderate/ severe (similar with Fazekas score)?

An additional column was added to correspond to LA severity

Probably the specification of the MRIs are given in the publications, was there a difference? At least it should be given in a sentence or even better in the table.

Column of MRI specification was added.

  • Discussion

-17 publications are mentioned, 19 were enrolled, probably typo.

It was corrected

-The summary is important, and relevant according to the review. Nevertheless, most of the studies are from Asia (13/19) /5 from Europe, 1 from America/. It is important to mention in the discussion part as well, since it is known that some aspects of stroke and risk factors may differ between these regions.

According to your suggestions, this was added and is marked with green:

“we acknowledge, however, that most study populations are from Asia and their results may not be generalizable to the world population.”

Thank for the precise guidance and the helpful input

Yours Sincerely,

Dr Tsiptsios

Reviewer 3 Report

The article provides a comprehensive review of the role of leukoaraiosis (LA) as a biomarker for stroke recurrence. The paper is well-structured, with a clear introduction, detailed results, and a thoughtful discussion. The topic is relevant and significant, as stroke recurrence is a critical concern in clinical practice. The study's approach to categorizing the findings based on stroke subtypes adds clarity to the discussion. Overall, the article is well-written and informative. Minor comments:

- Including a section on future research directions would be beneficial. Highlighting areas where additional research is needed (including preclinical models of LA) and potential study designs to address existing gaps could guide future researchers and encourage further investigation in this area.

- While the paper acknowledges some limitations, such as excluding certain stroke populations and the lack of investigation into sociodemographic characteristics, it would be helpful to discuss any potential sources of bias in the individual studies included in the review.

Author Response

Dear Reviewer 3:

Many thanks for your prompt response and your time spent reviewing our manuscript.

Your comments were thoroughly investigated and appropriate modifications were made to the text as follows:

- Including a section on future research directions would be beneficial. Highlighting areas where additional research is needed (including preclinical models of LA) and potential study designs to address existing gaps could guide future researchers and encourage further investigation in this area.

According to your suggestion changes were made and are highlighted with yellow:

“To uncover the processes underlying WMH dynamics, future investigators should examine microstructure in areas of WMH development using precise cerebral and systemic vascular function tests”

- While the paper acknowledges some limitations, such as excluding certain stroke populations and the lack of investigation into sociodemographic characteristics, it would be helpful to discuss any potential sources of bias in the individual studies included in the review.

According to your suggestion changes were made and are highlighted with yellow:

“Although we made every effort to avoid personal bias, we cannot exclude the possibility that some degree of bias may be present in our review. “

Thank you for the precise guidance and helpful input

Yours Sincerely,

Dr Tsiptsios